# Core-Shell Nanofibers of Polyvinylidene Fluoride-based Nanocomposites as Piezoelectric Nanogenerators

**DOI:** 10.3390/polym12102344

**Published:** 2020-10-13

**Authors:** Deepalekshmi Ponnamma, Mariem Mohammed Chamakh, Abdulrhman Mohmmed Alahzm, Nisa Salim, Nishar Hameed, Mariam Al Ali AlMaadeed

**Affiliations:** 1Center for Advanced Materials, Qatar University, Doha P.O. Box 2713, Qatar; mchamakh@qu.edu.qa; 2Department of Chemistry and Earth Sciences, College of Arts and Science, Qatar University, Doha P.O. Box 2713, Qatar; aa1601257@student.qu.edu.qa; 3Faculty of Science, Engineering and Technology, Swinburne University of Technology, Hawthorn, Melbourne, VIC 3122, Australia; nsalim@swin.edu.au (N.S.); nisharhameed@swin.edu.au (N.H.); 4Materials Science & Technology Program (MATS), College of Arts & Sciences, Qatar University, Doha 2713, Qatar; m.alali@qu.edu.qa

**Keywords:** coaxial spinning, nanogenerator, semiconducting nanoparticles, hydrothermal synthesis

## Abstract

Flexible piezoelectric nanogenerators (PENG) are widely applied to harvest sustainable energy from multiple energy sources. The rational and simple design of PENG have great potential in soft electronics. Here we design a highly flexible PENG using the polyvinylidene fluoride (PVDF) and its copolymer, polyvinylidene hexafluoropropylene (PVDF-HFP) with two nanoarchitectures of semiconducting metal oxides, TiO_2_ and ZnO. The nanotubes of TiO_2_ and nanoflowers of ZnO are embedded in these different polymeric media by solvent mixing, and new fiber mats are generated by coaxial electrospinning technique. This process aligns the dipoles of polymers and nanomaterials, which is normally a pre-requisite for higher piezo potential. With excellent mechanical strength and flexibility, the tailored lightweight fiber mats are capable of producing good output voltage (a maximum of 14 V) during different mechanical vibrations at various frequencies and in response to human motions. The hybrid nanocomposite PENG is durable and inexpensive and has possible applications in wearable electronics.

## 1. Introduction

Recently, the internet of things and Industry 4.0 [1] have been an inevitable part of human life. Portable electronic devices such as actuators, smart phones, displays and sensors are essential parts of technology [2]. As many of these technologies demand power sources for their continuous and long-term operations, self-powering devices with good flexibility, mechanical robustness, highly efficient energy storage performance and environmental friendliness are highly required in this new era [3]. Harvesting energy from ambient sources such as acoustic waves or airflow is well investigated because of its sustainable and reliable applications [4]. Mechanical motion or thermal fluctuations are being transformed to electrical energy output by the piezoelectric or pyroelectric polymers and its composites. Numerous self-powering nanodevices and sensors work on a piezoelectric energy scavenging mechanism, where mechanical energy is converted to electrical voltage [5,6,7]. Using polymers offers flexibility and durability to the nanogenerators and at the same time maintains low cost. Ceramic composites of lead zirconate titanate (PZT), barium titanate (BaTiO_3_), etc. have remarkable energy harvesting performance coupled with magneto electric and dielectric properties [8,9]. However, demand for flexible and wearable nanogenerators in electronics opens up the scope of finding more useful alternatives for piezoelectric applications. In this regard, polymer-based materials are remarkable substitutes [10]. The piezoelectric devices on flexible substrates prevent the breaking of piezoelectric nanoparticles within, and the whole device can be used in flexible and wearable electronics [11]. 

The major characteristics of polyvinylidene fluoride (PVDF) and its copolymers include low cost, mechanical flexibility, low temperature processing and ferroelectricity. Among the various phases of PVDF, α phase is the thermodynamically stable one, but it is nonpolar [12,13]. The crystalline β phase is responsible for a higher ferroelectric piezoelectric response, and the conversion from α to β is highly preferred. Processes such as poling, annealing and doping with nanoparticles such as ZnO, BaTiO_3_, etc. can enhance the α to β transformation and thus improve the piezo performance [14]. The conducting fillers in higher percentages also cause the increase of the leakage current, and in such cases, more focus should be given to enhance the piezoelectric voltage generation by reducing the leakage current [15]. Electrospinning is a prominent fiber manufacturing technique to develop piezoelectric nanocomposite fibers as the process induces dipole alignment. Such electrospun nanofibers are widely applied in manufacturing membranes for filtration, separators in batteries, scaffolds in tissue engineering, sensors in electronics and in wearable technologies [16,17,18]. While small pore sizes and good permeability of these light and flexible membranes add to their numerous applications, many nanoparticles are incorporated to the neat polymers to get highly efficient nanofibers for different applications [19,20]. The ferroelectric and semicrystalline PVDF possesses excellent mechanical and thermal properties, good chemical stability, superior antioxidation and highly efficient mechanical energy harvesting. The electronegativity difference of F and H/C atoms within its skeleton causes a strong electric dipolemoment, which is the cause for its remarkable electroactive properties [13,14]. 

Baniasadi et al. developed stretchable yarns of PVDF-TrFE and the polymer coils made out of those yarns stretched up to 740% strain [21]. The piezoelectric properties of the yarns were tested both by direct and converse methods. An energy to failure of up to 98 J/g was achieved by the yarns with high mechanical strength and roughness. Metal oxide nanoparticles of nickel oxide (NiO), zinc oxide (ZnO), nickel ferrite (NiFe_2_O_4_), etc. enhance the β phase content of PVDF and thus the piezoelectric output performance of the composite [17,20,22]. ZnO nanowires are reported for their piezopotential along the c-axis when subjected to a compression, tension or bending movement [23]. However, the possibility of agglomeration of the particles normally limits the complete utilization of the piezo performance. To enhance the dispersibility of the ZnO in PVDF media, SiO_2_ particles were introduced to the polymer medium and a hybrid nanoarchitecture was created by Dutta et al. [3]. A mechanical energy harvester with a tactile electronic sensor was achieved based on the NiO@SiO_2_/PVDF nanocomposite. Another semiconducting metal oxide, TiO_2_, was also applied in fabricating piezoelectric polymer nanocomposites [24], as it suppresses the tunneling current of polymer composites and thus helps in enhancing the breakdown strength. 

Our group investigated the influence of the electrospinning process on determining the piezoelectric output voltage of various nanocomposites and its influence on the dipole alignment [12,16,18,19]. Core shell nanofibers with two piezoelectric polymers at the inner and outer parts are systems of much interest because of the special fiber morphology. Coaxial electrospinning provides wider possibilities because of the efficiency of the method, adjustable core sheath dimension (thickness) and large varieties of material combinations [25]. Though fabrication of nanofibers by coaxial electrospinning has been reported for several applications, particularly in drug delivery and tissue engineering [26], the mechanical energy harvesting has not been addressed. In this work, two polymers, PVDF-HFP and PVDF, are used to coaxially electrospin the fiber mats with core-shell morphology. Electrospinning not only helps in the alignment of the dipole and crystallization effect, but also polarizes the inorganic filler materials and induce inductive charges. The coaxial electrospinning particularly introduces the synergistic filler-matrix reinforcing effect, and the current results shows that the PVDF-HFP/PVDF (core/shell) containing ZnO and TiO_2_ nanoparticles exhibits a maximum output voltage of 14 V. In addition, the nanoparticles present in the inner core and outer shell enhance the crystalline property of the polymer fibers and reduce the breakdown strength. The core shell fibers also achieved piezoelectric voltage generation in response to human motion.

## 2. Experimental Techniques

### 2.1. Materials

PVDF pellets (M_w_ = 180,000), PVDF-HFP pellets (M_w_ = 440,000), N,N dimethylformamide (DMF) and acetone (both of purity ≥ 99.8%) were obtained from Sigma Aldrich (Doha, Qatar). The reagents used for the nanoparticle preparation, zinc acetate dihydrate [Zn(CH_3_COO)_2_.2H_2_O], monoethanolamine or MEA (C_2_H_7_NO), polyethylene glycol (PEG), ethanol (purity ≥ 99.8%), TiO_2_ nanopowder, etc. were also obtained from Sigma Aldrich (Doha, Qatar) and used without further purification. 

### 2.2. Synthesis of TiO_2_ and ZnO Nanomaterials

Both TiO_2_ and ZnO nanomaterials were synthesized by hydrothermal methods. This method allows for the formation of TiO_2_ and ZnO in specific tubular- and flower-like morphologies, which influence the rate of dispersion within the polymer and thus the property enhancement. In addition, the crystalline geometries in terms of lattice spacing affect the piezoelectric activity of the nanogenerators as well. During the synthesis of TiO_2_, a mixture of 2.4 g anatase TiO_2_ powder and 40 mL 10 N NaOH were dispersed very well in a Teflon beaker by vigorous shaking. This mixture was then transferred to a preheated autoclave at 135 °C and kept in the oven at that temperature for 10 h. After the complete growth of TiO_2_, the precipitate was filtered and washed in water and 0.1 N HCl until neutral pH. The powder was thereafter dried at 80 °C for 4 h in an oven and used for further analysis.

For the synthesis of ZnO, a specific amount of zinc acetate dihydrate was dissolved in 50 mL distilled water by magnetic stirring. Then, 0.5 g PEG surfactant was added to this solution followed by 3 mL MEA. After magnetic stirring for 30 min, the solution was transferred to a Teflon-capped autoclave and kept in an oven at 140 °C for 15 min. The autoclave was cooled after the reaction, and the precipitate generated was washed with distilled water and ethanol several times. The precipitate was dried at 80 °C in a hot air oven for 12 h and then annealed in a tube furnace at 400 °C for 2 h to obtain ZnO. 

### 2.3. Fabrication of Nanocomposite Membranes by Coaxial Electrospinning

The PVDF and PVDF-HFP nanocomposites containing ZnO and TiO_2_ nanomaterials were synthesized by a coaxial electrospinning process, as schematically represented in Figure 1. For this, separate dispersions of PVDF and PVDF-HFP were made by dissolving 2 g pellets of the polymer in 15 mL solvent mixture of DMF and Acetone (1:1). The polymers were dissolved by the magnetic stirring process for 2 h at 70 °C. Both nanomaterials (TiO_2_ and ZnO) were separately dissolved in 5 mL of the same solvent mixture by ultrasonication for 60 min. Later, these dispersions were added to the respective polymer solutions to generate various nanocomposite solutions of PVDF and PVDF-HFP. Coaxial electrospinning follows the same principle of electrospinning (Taylor Cone formation with applied voltage and thereafter fiber formation), but two solutions are fed to the needle through outer and inner tubes [27,28]. These inner and outer solutions are represented as core and shell polymers that are, respectively, fixed as PVDF-HFP and PVDF. The nanomaterials, ZnO and TiO_2_, are added to the polymers in such a way to generate samples as shown in Table 1, and the hybrid effects of nanomaterials are investigated by reinforcing both polymers with nanomaterials.

### 2.4. Characterization Techniques

Surface morphology analysis was carried out using scanning electron microscope (Nova Nano SEM 450, Deutschland, Germany) and the morphology of the nanofillers was investigated using transmission electron microscope (TEM) (Phillips CM12). Structural properties of the coaxial fibers were investigated by X-ray diffractometer (Empyrean, Panalytical, Malvern, UK) within the diffraction angle (2θ) range 10° to 70° and Fourier transform infrared spectrophotometer (FTIR, PerkinElmer Spectrum 400, Waltham, MA, USA) during 2000–400 cm^−1^ wavenumber with a resolution of 2 cm^−1^. Thermogravimetric analysis (TGA) was performed for the coaxial fibers using perkin Elmer Pyris 6 TGA, from 30 to 600 °C at a heating rate of 10 °C/min. Differential scanning calorimeter (DSC 8500 PerkinElmer, Waltham, MA, USA) was used to measure the melting and the crystallization behavior at a temperature range of 20 to 200 °C at 10 °C/min. Finally, the piezoelectric studies were done by a specific assembled set-up consisting of a frequency generator, amplifier, vibrating shaker, resistance box and data acquisition system, as per the previously established protocol [20,22]. Samples were cut into rectangular pieces of 2 × 2 dimension and attached to conducting wires from both sides using the silver paste and then covered with aluminum foil. Around 2.5 N force was applied on the sample (kept above the shaker) in a longitudinal direction. Frequencies are adjusted in the frequency generator (Keysight 33220A function arbitrary waveform generator, Aurora, OH, USA) from 15 to 50 Hz, which was amplified (amplifier Crown XTi 2002 series, Aurora, OH, USA) and then sent to the shaker (VTS vibration test systems). While the shaker and samples on top of it vibrates with the applied frequency and force applied on the sample, voltage was generated in response to it. A resistance box was also in the circuit with a fixed resistance value of 1 MΩ. The voltage and current values are obtained from the computer connected with the set up through signal express programming. Dielectric properties of the coaxial fibers of 2 cm diameter during 10^−1^–10^7^ Hz frequency were tested using the broadband dielectric spectroscope Alpha-T (Novo control Technologies, Deutschland, Germany) under room temperature. 

## 3. Results and Discussion

### 3.1. Morphology of TiO_2_ and ZnO Nanomaterials

Figure 2a presents the schematic illustration for the synthesis of both TiO_2_ and ZnO nanomaterials. The hydrothermal process was adopted for the synthesis of both nanomaterials and different morphologies are obtained. Hydrothermal reaction for 10 h transforms the TiO_2_ spherical particles to sheets as it ruptures and thereafter rolling of layers in specific direction takes place to form the tubes [29]. While TiO_2_ achieved tubular shape, ZnO possessed a flower-like appearance after the hydrothermal treatment. This happens when zinc acetate reacts with PEG and MEA to develop ZnO nanorods, and then the nanorods grow in situ to flowers by the Ostwald ripening mechanism [29]. These different morphologies are clear from the SEM images of Figure 2b,c. As shown in Figure 2b, TiO_2_ nanotubes show fibrous morphology [30]. During hydrothermal treatment, the spherical particles of TiO_2_ collapse and reform to nanotube morphology [31] which is further clear from the TEM image (Figure 2d). The TiO_2_ nanotubes are well distributed without aggregation and the average diameter of the tube was observed to be 20–30 nm. ZnO nanostructure is observed through the SEM image (Figure 2c) and the TEM image (Figure 2e). In fact, the MEA reagent added during the ZnO synthesis acts as a triggering agent towards the formation of flower-like appearance [20,22]. The MEA generates seed nuclei, which grows to longer rods and finally connects together in specific fashion to form the flowers, according to the Ostwald ripening mechanism [32].

### 3.2. Morphology and Structure of Coaxially Electrospun PVDF-HFP/PVDF Fibers Containing TiO_2_ and ZnO Nanomaterials

Figure 3 shows the morphology of the spun fibers. Perfect and well-distributed fibers are formed in all composites without the formation of beads or agglomerations. The influence of TiO_2_ and ZnO nanoparticles on the core shell morphology of the fibers is very clear from the SEM images. In all cases, the average fiber diameters were calculated and included in the Figure 3 as insets. The average fiber diameter values show a wide variation when compared between the different nanocomposite systems [33,34]. It is observed that the neat PVDF-HFP/PVDF (PH-PF) coaxial fiber possesses an average diameter of 268 ± 102 nm, which enhanced with the ZnO addition. However, when TiO_2_ nanotubes are added, the fiber diameter increased to 418 ± 155 nm. This deviation in the average fiber diameter between the ZnO and TiO_2_ nanoparticles reinforced composites can be attributed to the difference in flow behavior with the different morphology of the nanomaterials [27,35]. The tubular TiO_2_ with 20–30 nm average diameter, may help in its well distribution within the PVDF medium, and this causes a good networking effect with comparatively low flowing rate with the core PVDF-HFP. This kind of increase in fiber diameter due to increased core feed rate is in well accordance with Nguyen et al. [36], who investigated the electrospun fiber morphology of PEG/PVDF with variable feed rates. Similar behavior was also reported for other core-shell nanofibers [37]. However, the PVDF-HFP/PVDF containing both nanoparticles in the inner and outer structure (Figure 3d) exhibited the least fiber diameter of 196 ± 76 nm. This illustrates the significance of reinforcing both the inner core and outer shell with different nanomaterials, which can make a perfect networking structure and introduce the shear thinning effect. Figure 3e,f show the hybrid composite nanofiber’s cross sectional view and the TEM image, respectively, which again confirms the fiber morphology, diameter and the coaxial structure with a comparatively thinner outer shell. This result well matches the reported works of Nguyen et al. [36].

Structural analyses of the samples were done using the X-ray diffraction studies and the FTIR spectra. Figure 4a represents the XRD pattern for all composite samples in addition to the TiO_2_ and ZnO nanoparticles. In the XRD pattern of composites, the peaks at 18.1 and 20.1 correspond to (100) and (110) reflections of α-PVDF and the peak at 20.8 is due to (111) plane of β-PVDF. The peaks that correspond to the nanoparticles are hidden in the pattern of the composites which represents the high-level distribution (trapping) of nanomaterials within the polymer chains [38]. XRD also identifies the crystallinity of the nanocomposites, which is associated with the formation of hydrogen bonds between the F atom of the polymer and the inherently present –OH groups on the filler surface. However, the higher concentrations of the filler, restrict the polymer chain mobility and enhance the stiffness, thus lowering the crystallinity [15]. The crystallinity and the interfacial interactions are addressed also by studying the peak intensity ratios. The ratio of peak intensities at 20.3 (characteristic of β phase) and 18.2 (α phase) is represented as I_20.3_/I_18.2_. This ratio changed between all the samples according to the following values, 1.46 (PH-PF), 1.54 (PH-PFT), 1.45 (PH-PFZ) and 1.61 (PHT-PFZ). An increase in this value indicates the capability of filler particles (especially the hybrid effect of TiO_2_ and ZnO) in transforming the α phase to the β phase. This α to β transformation is evidence of increasing the crystallinity of the polymer, which again enhances the piezoelectric output performance [39]. 

In the FTIR spectra of the coaxial electrospun fibers (Figure 4b), α-PVDF is responsible for the peaks at 614, 763, 795, 853, 974, 1071 and 1172 cm^−1^ and the β-PVDF causes the absorbance peaks at 879 and 840 cm^−1^. The presence of β-phase can be calculated from the FTIR spectral details by the following formula [12]: (1)$$F\left(\beta \right)=\frac{A_{\beta }}{\left(1.26A_{\alpha }+A_{\beta }\right)}$$
where *A_α_* and *A_β_*, respectively, represent the absorbance at 840 and 763 cm^−1^.

Here the *F*(*β*) values of the composites are 0.131 (PH-PF), 0.135 (PH-PFT), 0.127 (PH-PFZ) and 0.155 (PHT-PFZ), which is in clear correlation with the crystallinity data from the XRD. The hybrid composite sample shows the highest α to β transformation, as expected. This indicates the influence of hybrid filler materials presented in the outer and inner layers of the core shell nanofibers [40]. In fact, the fingerprints for the β phase of PVDF are the peaks at 877 and 1273 cm^−1^. The peak at 1173 cm^−1^ is associated with –CF_2_ stretching vibrations due to the ZnO/–CF_2_ interactions and thus the effective mass loss of –CF_2_ dipoles.

### 3.3. Crystallinity Studies and Thermal Characteristics of Coaxially Electrospun PVDF-HFP/PVDF Nanofibers

DSC analysis shows the crystallization kinetics for the coaxially electrospun PVDF-HFP/PVDF nanocomposite fibers (Figure 5). All composites show different peak positions at different temperatures, which represents the different melting temperature and crystal morphology in all cases. All graphs show the pronounced effect of the hybrid coaxial nanofiber. Figure 5a shows the cooling curves and the crystallization temperatures for all fibers at 152.06, 149.06, 150.72 and 141.29 °C, respectively for PH-PF, PH-PFT, PH-PFZ and PHT-PFZ. The values show a regular decrease, illustrating the influence of nanomaterials in immobilizing the polymer, or in other words, this is a good indication of filler networking effect [41]. As per Figure 5b, the melting peaks for all samples PH-PF, PH-PFT, PH-PFZ and PHT-PFZ are at 177.06, 176.37, 176.05 and 175.45 °C, respectively. The PHT-PFZ sample also shows an additional shoulder at 169.89 °C. The melting peaks also show similar trend of decrease as in crystallization temperatures, further evidencing the filler distribution.

When the sample PVDFHFP-TiO_2_/PVDF-ZnO was treated at different heating/cooling rates, interesting results are revealed, as per Figure 5c,d. The values of crystallization temperature for the hybrid coaxial nanofiber were, respectively, 149.80, 141.29, 136.12 and 133.56 °C at 5, 10, 20 and 30 °C/min. The poor affinity between the hydrophobic polymers (both PVDF and PVDF-HFP) and the hydrophilic filler particles (ZnO and TiO_2_) can reduce the glass transition and thus enhance the chain mobility. This reduces the energy barrier needed for polarization switching and triggers the fast polarization, and thus reduces the coercive field [15]. While the melting temperatures first decreases as the rate enhanced from 5 to 20 °C/min with the values of 176.74 °C at 5 °C/min, 175.45 °C at 10 °C/min, 172.46 °C at 20 °C/min, an increase was also observed at 30 °C/min (174.85 °C). In all cases, the peak width is comparatively increasing, as there is a time lag between the sample and the reference in achieving the thermal equilibrium [42,43]. This variations in crystallization and melting temperatures with rate of cooling/heating has strong effect on the phase transition and crystallization kinetics.

The thermal characteristics of the coaxial nanofibers are provided in Appendix A. It is clear that the thermal degradation temperature and thermal stability are decreasing with the addition of nanomaterials. The onset of degradation is lower for composites containing TiO_2_ nanomaterials, but for the ZnO containing nanocomposite fibers, the degradation value is higher. The final weight residue increased with the filler reinforcement, which indicates the improved cyclization of the polymer, enhancing the heat resistant molecular structures and favoring carbonization [42]. 

### 3.4. Energy Harvesting Performance of Coaxially Electrospun PVDF-HFP/PVDF Nanocomposite Fibers 

Han and Steckle [25] recently reviewed the significance of coaxial electrospinning in generating energy and the significance of such fibers in smart textiles. Pan et al. [44] generated PVDF hollow fibers by coaxial electrospinning, and achieved a piezoelectric output voltage, 2.46 times larger than that of solid PVDF fibers. Figure 6 represents the piezoelectric output voltage values obtained for the coaxial electrospun fibers based on PVDF and PVDF-HFP. During piezoelectric analysis, the strain of the sample creates crystal structure deformation inducing charges towards the piezo potential. Accordingly, external charges flow to the capacitor and develop an output signal. When the strain is removed, the potential diminishes, and free charges flow back due to the electric pulse in opposite direction. Thus, the positive and negative values indicate the application and withdrawal of external pressure to the sample [15]. Long-term application of the piezoelectric device definitely depends on the robustness of the material, and for this reason devices performance towards multiple strain cycles is analyzed. 

It is clear from the figure that 45 Hz is the resonance frequency for most of the samples. If the results are compared at this particular frequency, the neat polymer shows an output performance of 1.75 V (peak to peak value) which slightly enhances to 2 V for the TiO_2_ containing fibers and shoots up with the presence of ZnO to 5.5 V. For the hybrid composite containing both TiO_2_ and ZnO the piezoelectric peak to peak output voltage reaches 14 V, and this could be due to the maximum polarization happening at this particular composition [19]. 

Comparatively higher piezoelectric output voltage for the ZnO and TiO_2_ containing nanocomposite fibers is in good connection with the crystallinity data obtained from the XRD, DSC and FTIR studies. The observed α to β transformation for the PVDF-HFP/PVDF polymeric phases by the nanomaterials is the major reason behind the enhancement in crystallinity and for the piezoelectric voltage generation. 

The output current values for these composites were measured in addition to the power density, as illustrated in Figure 7. As Figure 7a compares the maximum output voltage of all samples, the core-shell fibers of PVDF-HFP/PVDF with both ZnO and TiO_2_ nanomaterials exhibits the highest output performance. The current density and power density (Equation (2)) for these samples were also measured from the average current values achieved at the maximum vibrational frequency and at specified resistance values.
(2)$$\mathrm{Power}\;\mathrm{Density}=\frac{\mathrm{Power}}{\mathrm{Volume}}=\frac{\mathrm{VI}}{\mathrm{At}}$$
where V and I are respective voltage and current across the load 1 MΩ. A and t are the respective area and thickness of the samples. The values illustrated in Figure 7b observes the comparatively higher performance for the hybrid core-shell nanofiber as expected from the crystallinity information. It is also noted that the PHT-PFZ shows maximum piezoelectric output performance as a result of the combined influence of PVDF crystallinity as well as the ZnO piezoelectricity [45]. The average fiber diameter for these fibers were also the least allowing proper dipolemoment orientation and good network formation, which also contributes towards the piezoelectric mechanism [20]. The real vibrations due to hand pressing and bending were done on the samples and the voltage responses achieved are given in Figure 7c,d, respectively. In both cases, the output voltage showed clear variation, attributed to the material’s ability in converting human mechanical energy. The insets of Figure 7c,d show the PHT-PFZ fibers with conducting wires attached on both sides by silver paste and covered in aluminum foil. The pressing and bending cycles show regular patterns as the graphs represent a very short time period. Noises were fully avoided for plotting the graphs. 

### 3.5. Dielectric Properties of Coaxially Electrospun PVDF-HFP/PVDF Fibers 

Figure 8 shows the variation of dielectric properties of the coaxial fibers with frequency and temperature change. As per Figure 8a, the dielectric constant increases with filler addition, and for the hybrid filler combination the value becomes the highest (about five times higher than the neat polymer). The enhanced dielectric constant with filler addition can be correlated with the Maxwell–Wagner–Sillars (MWS) interfacial polarization effect [46]. According to this effect, accumulation of space charges and short-range dipole–dipole interactions occur at the polymer-filler interface and cause interfacial polarization and thus increase the dielectric constant. In addition to the MWS effect, the enhanced electroactive β phase content in the nanocomposite contributes to the enhancement of dielectric constant. With frequency enhancement, the dielectric constant shows a sharp decrease for the hybrid composite compared to the neat core shell fiber due to its stronger frequency dependence and networking effect [47]. This reduction in dielectric constant is due to the low number of aligned dipoles. At higher frequency, the dipoles did not get enough time to orient and so they lag to the field.

Variation of tan δ with frequency in Figure 8b further reveals the polarization effects within the nanocomposite fibers. The inset graph highlights the V-shaped curves for tan δ versus frequency. The dielectric loss mainly includes the conductance loss and polarization loss, respectively, at the 10^4^ and 10^4^–10^6^ frequencies. While the conductance loss decreases with frequency at lower frequencies, the change in electric field and the corresponding hysteresis generates polarization loss at higher frequencies. This is the reason for the V-shaped curves of tan δ versus frequency plot [47]. The significant reduction in dielectric loss for the PH-PFT composite can be due to the interactions of TiO_2_ with the –CH_2_ groups of PVDF, which hinders the flipping of dielectric constant. However, for the composite fibers containing the ZnO, equal distribution of nanomaterials generates networking influence and creates a well-distributed fiber structure. The frequency at which lowest tan δ value is observed is the relaxation time. From the inset of Figure 8b, it is clear that the relaxation time decreases for the PHT-PFZ when compared to the neat polymer. This correlates with the networking structure as the polymer chains are restricted for movement when finely distributed particles exist in nanocomposites.

The variation of ε′ and tan δ of the coaxial electrospun fibers against temperature (−90 to 120 °C) at 10 Hz frequency is given in Figure 8c,d, respectively. With temperature, the ε′ gradually increases confirming the dipolar polarization existing in the samples. For the composites, the values are higher due to the enhancement in β-phase content and thus the polarization due to the presence of nanomaterials [10]. The filler particles also generate interfacial polarization, which act as trapping centers and enhance the ε′ again. A sudden change in the dielectric constant value is observed during 75–85 °C, which can be attributed to the Curie temperature of the polymer nanocomposites. In addition, there is a slight variation in these temperature values among all composites, indicating the nanoparticle induced polarization occurring in the samples [48]. 

Finally, the dielectric properties of the composites are also in good connection with the piezoelectric output voltage and crystallinity data, as the hybrid core-shell nanocomposite fiber showed the best correlation. It is believed that the dipole alignment, crystallinity, α to β phase transformation within the polymer and the interfacial polarization are induced by the presence of ZnO and TiO_2_ nanomaterials, thus making it useful in harvesting mechanical vibrations (including human motion).

## 4. Conclusions

In summary, we have tailored new nanogenerators based on the PVDF-HFP/PVDF (core/shell) coaxial nanofibers containing ZnO and TiO_2_ nanoarchitectures. Core-shell fibers are obtained and the influence of filler particles on the inner core and outer shell of the nanofibers are studied. The generated nanofibers produced a maximum peak to peak output voltage of 14 V, indicating the good mechanical to electrical voltage conversion efficiency. This result was obtained for a circular sample weighing 0.1 g, and it evidences the maximum efficiency and simplicity in design. In addition, the nanofibers produced remarkable energy storage performance with five times higher dielectric constant when compared to the neat polymer.

## Figures and Tables

**Figure 1 polymers-12-02344-f001:**
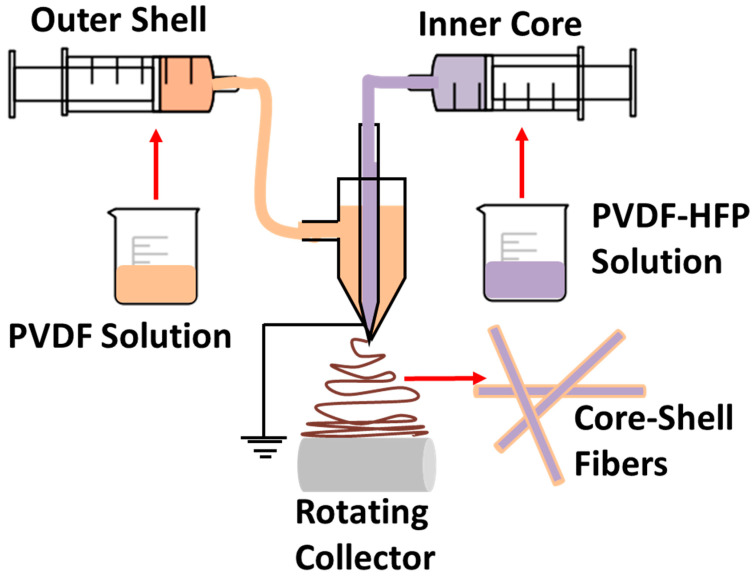
Schematic illustration of the coaxial electrospinning process.

**Figure 2 polymers-12-02344-f002:**
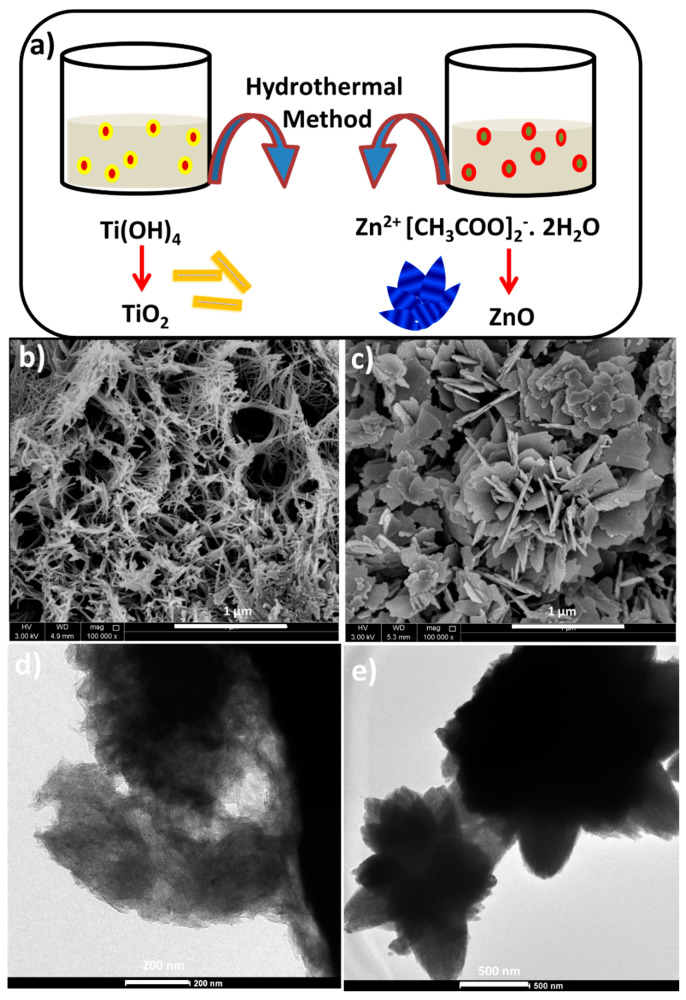
(**a**) Schematic representation of TiO_2_ and ZnO preparation. SEM images of (**b**) TiO_2_ and (**c**) ZnO. TEM images of (**d**) TiO_2_ and (**e**) ZnO.

**Figure 3 polymers-12-02344-f003:**
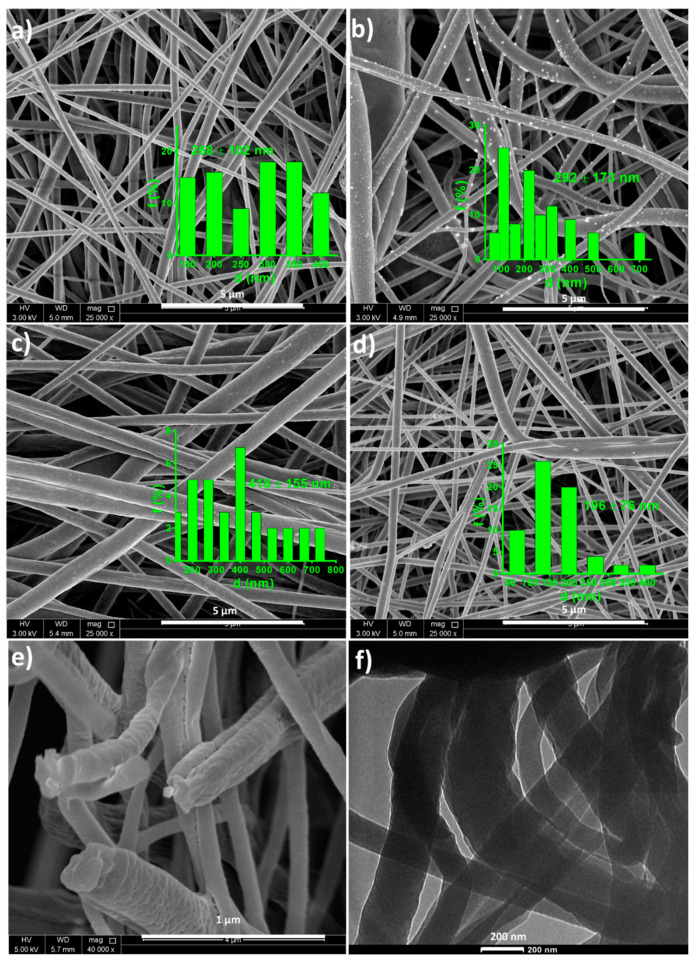
SEM images of (**a**) PH-PF (**b**) PH-PFZ (**c**) PH-PFT (**d**) PHT-PFZ composite fibers, (**e**) PHT-PFZ cross sectional SEM image and (**f**) TEM image of PHT-PFZ fibers.

**Figure 4 polymers-12-02344-f004:**
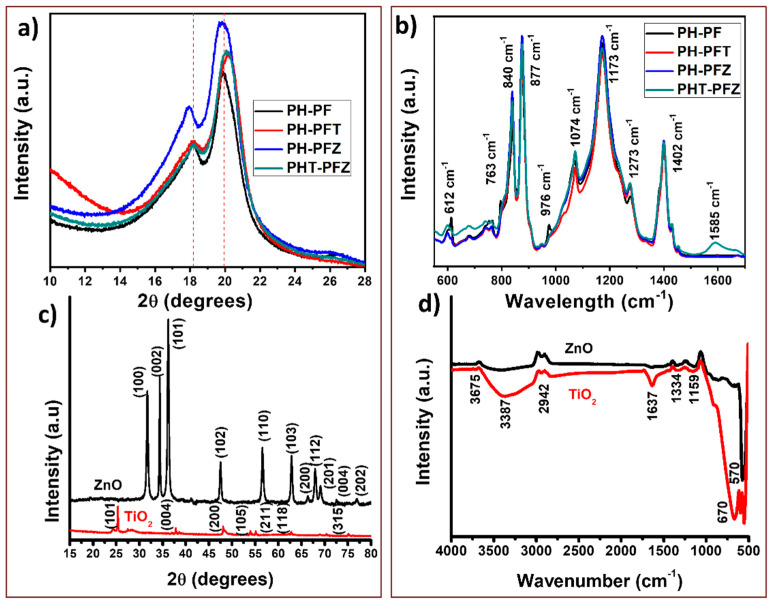
XRD patterns and FTIR spectra of (**a**) the PVDF-HFP/PVDF composites and (**b**) the nanomaterials, (**c**,**d**) respectively show the XRD and FTIR spectra of ZnO and TiO_2_.

**Figure 5 polymers-12-02344-f005:**
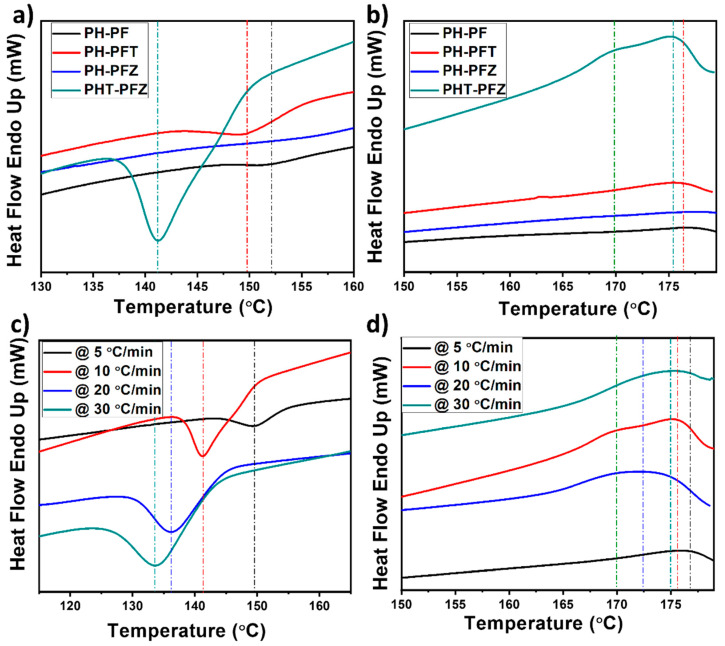
Differential scanning calorimeter (DSC) curves for the coaxial electrospun fibers (**a**) endothermic and (**b**) exothermic heat flow curves for the nanocomposites (**c**,**d**) heat flow behavior of the hybrid PHT-PFZ composite at different heating/cooling rates.

**Figure 6 polymers-12-02344-f006:**
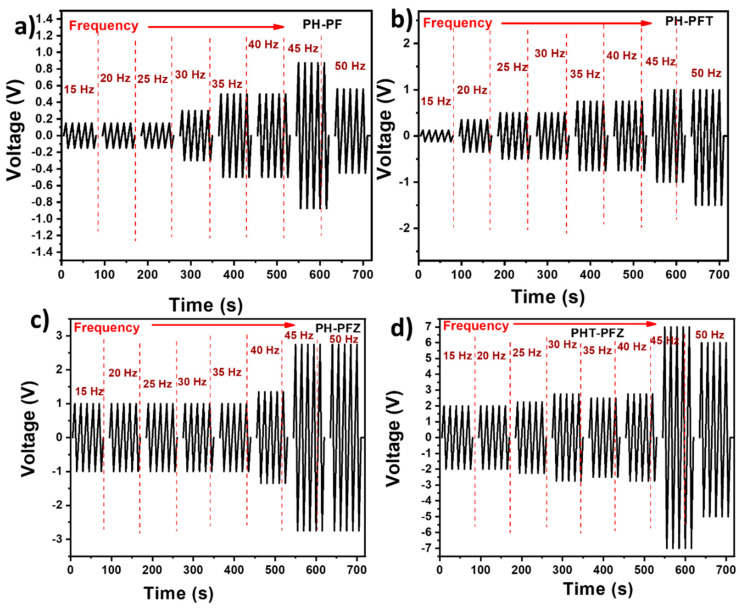
Piezoelectric output voltages for the coaxial electrospun fibers (**a**) PH-PF (**b**) PH-PFT (**c**) PH-PFZ and (**d**) PHT-PFZ.

**Figure 7 polymers-12-02344-f007:**
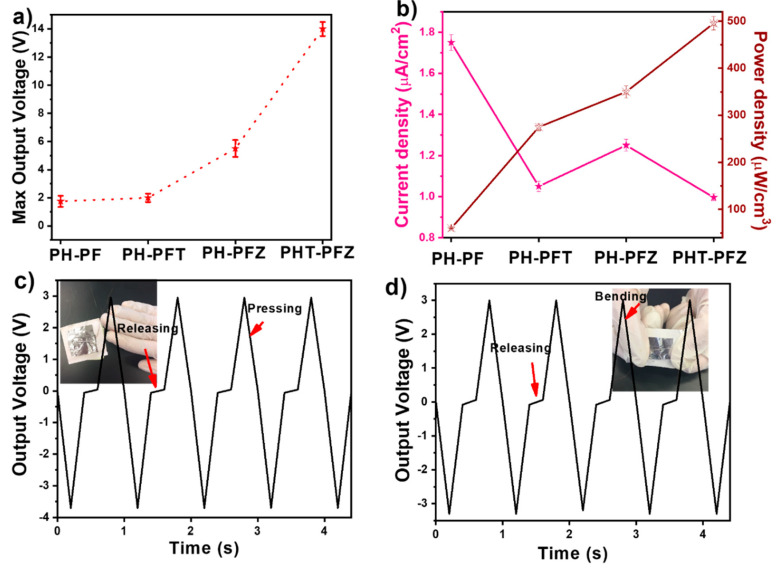
(**a**) Maximum peak to peak output voltage (**b**) current density and power density; voltage responses of the samples to (**c**) hand pressing and (**d**) bending motions (inset shows PHT-PFZ fibers with conducting wires attached on both sides).

**Figure 8 polymers-12-02344-f008:**
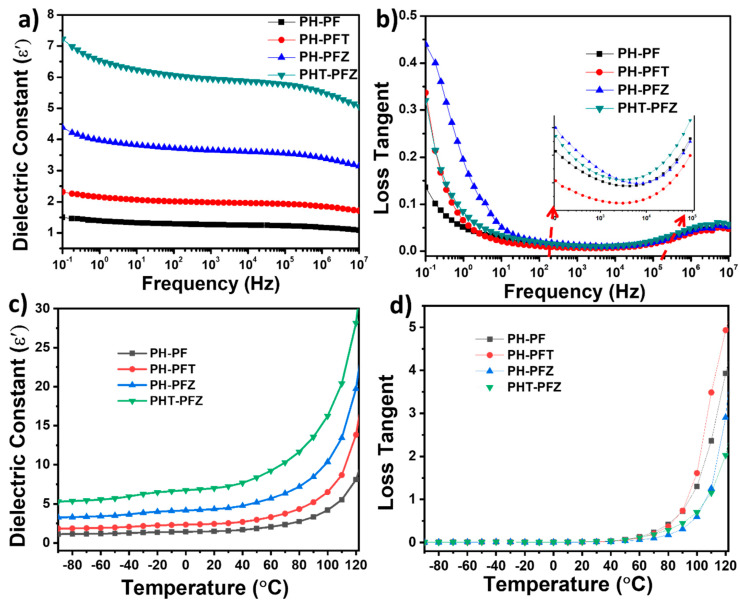
Dielectric properties of the coaxial electrospun fibers (**a**) dielectric constant and (**b**) loss tangent with frequency at room temperature; (**c**) dielectric constant and (**d**) loss tangent variation with temperature at 10 Hz.

**Table 1 polymers-12-02344-t001:** Details of the samples fabricated.

Samples	Inner Core	Outer Shell	Fiber Diameter (nm)	Crystallization Temperature (°C)
PH-PF	PVDFHFP	PVDF	268 ± 102	152.06
PH-PFZ	PVDFHFP	PVDF-ZnO	292 ± 173	150.72
PH-PFT	PVDFHFP	PVDF-TiO_2_	418 ± 155	149.06
PHT-PFZ	PVDFHFP-TiO_2_	PVDF-ZnO	196 ± 76	141.29

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
