# Peer review of "Core-Shell Nanofibers of Polyvinylidene Fluoride-based Nanocomposites as Piezoelectric Nanogenerators"

_polymers, 2020, doi:10.3390/polym12102344_

Round 1

Reviewer 1 Report

In this work the authors the authors study a PVDF based piezoelectric energy harvester. The harvester is fabricated using electrospinning process and TiO2 and ZnO is added to the PVDF to achieve composite material. The authors argue with experimental results that the composite material has better performance.

I would like to recommend this work for publication after some corrections.

  1. In section 3.4, the reported values of voltage are peak-to-peak values which might confuse some readers. In order to avoid this confusion, the authors should state this clearly in the manuscript. Also please provide formula used for power density calculation and specify the value of resistance. Finally, in figure 7b, the current for PHT-PFZ is lowest whereas the power is highest. How is this possible?
  2. The authors insist that the hybrid composite material PHT-PFZ has greatest beta-PVDF compared to other materials resulting is its highest output. However, in Section 3.2 PH-PFT has greater beta-PVDF than PH-PFZ, whereas in Section 3.4 PH-PFZ has greater output than PH-PFT. How can the authors explain this? Is it because ZnO is known to have piezoelectric behavior (REF: Choi, Min‐Yeol, et al. "Mechanically powered transparent flexible charge‐generating nanodevices with piezoelectric ZnO nanorods." Advanced Materials21.21 (2009): 2185-2189.)?
  1. Minor corrections:
    1.  Section 3.1, line 153, ZnO …SEM image (Figure 1c) and the TEM image (Figure 1e).
    2.  Section 3.2, title: TiO2
    3.  Section 3.2, line 191, Figure 3e and 3f show the hybrid composite… cross section and TEM.
    4.  Figure 3 caption. (e) and (f) are missing.

Author Response

Reviewer 1

Comments and Suggestions for Authors

In this work the authors the authors study a PVDF based piezoelectric energy harvester. The harvester is fabricated using electrospinning process and TiO2 and ZnO is added to the PVDF to achieve composite material. The authors argue with experimental results that the composite material has better performance.

I would like to recommend this work for publication after some corrections.

  1. In section 3.4, the reported values of voltage are peak-to-peak values which might confuse some readers. In order to avoid this confusion, the authors should state this clearly in the manuscript. Also please provide formula used for power density calculation and specify the value of resistance. Finally, in figure 7b, the current for PHT-PFZ is lowest whereas the power is highest. How is this possible?

=˃ The equation for power density is included in the revised manuscript. From the formula it is clear that the power density enhances with output voltage. In the case of PHT-PFZ the putput voltage is much higher and this is the reason why the power density is higher.

  1. The authors insist that the hybrid composite material PHT-PFZ has greatest beta-PVDF compared to other materials resulting is its highest output. However, in Section 3.2 PH-PFT has greater beta-PVDF than PH-PFZ, whereas in Section 3.4 PH-PFZ has greater output than PH-PFT. How can the authors explain this? Is it because ZnO is known to have piezoelectric behavior (REF: Choi, Min‐Yeol, et al. "Mechanically powered transparent flexible charge‐generating nanodevices with piezoelectric ZnO nanorods." Advanced Materials21.21 (2009): 2185-2189.)?

=˃ Yes, the reason for higher piezoelectric property in PHT-PFZ is also attributed to the effect of ZnO as mentioned by the reviewer. This is stated in the revised manuscript with back reference.

  1. Minor corrections:
  1.  
    1.  Section 3.1, line 153, ZnO …SEM image (Figure 1c) and the TEM image (Figure 1e).

=˃ Corrected

    1.  Section 3.2, title: TiO2

=˃ Corrected

    1.  Section 3.2, line 191, Figure 3e and 3f show the hybrid composite… cross section and TEM.

=˃ Corrected

    1.  Figure 3 caption. (e) and (f) are missing.

=˃ Corrected

Reviewer 2 Report

Overview:

Polymer ferroelectrics are very promising materials for use in low-cost wearable sensors and piezoelectric generators. The paper by Ponnamma et al. is dedicated to production and investigation of PVDF and PVDF-HFP -based core-shell fibers doped with ZnO and TiO2 nano particles, as well as voltage generating tissues made of the materials. While the topic has significance and may be interesting for readers, the manuscript not free from ambiguities, thus I recommend reconsidering the publication after major revisions.

Here I kindly ask to specify next several points in the manuscript:

  • I cannot agree with statement in line 43 that an application of ferroelectric ceramics is limited by their low chemical stability or thermal expansion. Many different types of ceramic-based (PZT) and single-crystal based (LiNbO3, PMN-PT) sensors, actuators and harvesters are suggested for almost any temperature range. The authors must give more detailed explanation or remove this statement.
  • Line 52: It seems to be there is missed words “…doped with”.
  • Line 98: The authors must explain or correct the strange order of commas in the molecular weights.
  • Line 102: The authors must specify the purity grades of the reagents that were used in the study.
  • Line 105: The authors must specify the role of hydrothermal process in synthesis of TiO2 and ZnO nanoparticles as the initial reagent also contained nanoparticles.
  • Line 141: The authors should check model of the dielectric spectroscope. There is no “GMbH” model in website of Novo Control Technologies.
  • The first paragraph of section 3.2 describing composite fabrication as well as Figure 2 must be in section 2 where materials preparation is specified.
  • The authors must give detailed explanation of chemical reactions given in Figure 2.
  • The authors must explain why flake-like nanoparticles of ZnO having size of several hundreds nm are not seen in SEM and TEM images given in Figure 3.
  • Descriptions of Figure 3(f), Figure 4 (two bottom plots) are missed in the captions to these figures.
  • Line 221: The authors mention absorbance at 838 cm-1 and 760 cm-1 in the text, but the same lines are marked as 840 cm-1 and 763 cm-1 in Figure 4. I think this discrepancy must be explained.
  • The peaks at 877 cm-1 1173 cm-1 have almost the same intensities for all the composites regardless there are or there are no inorganic nanoparticles. As the authors associate these peaks with β-phase and ZnO interaction with polymer chains, it must be explained why the undoped PH-PF possess the same magnitude and shape of these peaks.
  • The authors should explain why for hybrid coaxial nanofibers that contain at least two different polymers (PVDF-HFP and PVDF) we can see only one (for each composite) melting peak in DSC graphs.
  • The legend in Figure 5 (d) must be shifted in order not to hide graph lines.
  • The authors must specify, what of the investigates composites is related to Figure 5 (c) and (d).
  • The authors must give a detailed description how piezoelectric tissues shown in photos in Figure 7 were produced of the synthesized nanofiber composites.
  • The authors must give a detailed description of electrical measurement procedures. First of all, the load resistance must be specified. Second, what equipment was used to register data of Figure 6 (if oscilloscope, please give model)? Third, how deformation (or other type of piezoelectric excitation) was done and what was the magnitude of this excitation? In the current version of the manuscript these electrical measurements are the weakest part of the study.
  • Why in Figure 6 voltage signals at all frequencies have the same period in time domain? If these plots are only schematic, I feel there is no need to give them in the paper, the resonance frequency just as numerical value will be enough.
  • The author must give information how power densities and output current values were measured and by what equipment.
  • Graphs in Figure 7 (c) and (d) are too regular for simple hand pressing and bending motions, the authors must explain it.
  • Figure 8: Authors must specify what temperature was at measurements in (a) and (b) plots and what frequency was used for measuring (c) and (d) plots.
  • There are no sudden changes typical for ferroelectric phase transition at 81°C in Figure 8. It seems to be that the transition is at temperatures out of given in Figure 8 range (maybe, near melting point, or what is referred to the melting in DSC graphs is phase transition?). The authors should explain their statement.

Author Response

Reviewer 2

Comments and Suggestions for Authors

Overview:

Polymer ferroelectrics are very promising materials for use in low-cost wearable sensors and piezoelectric generators. The paper by Ponnamma et al. is dedicated to production and investigation of PVDF and PVDF-HFP -based core-shell fibers doped with ZnO and TiO2 nano particles, as well as voltage generating tissues made of the materials. While the topic has significance and may be interesting for readers, the manuscript not free from ambiguities, thus I recommend reconsidering the publication after major revisions.

Here I kindly ask to specify next several points in the manuscript:

  • I cannot agree with statement in line 43 that an application of ferroelectric ceramics is limited by their low chemical stability or thermal expansion. Many different types of ceramic-based (PZT) and single-crystal based (LiNbO3, PMN-PT) sensors, actuators and harvesters are suggested for almost any temperature range. The authors must give more detailed explanation or remove this statement.

=˃ The sentence is removed.

  • Line 52: It seems to be there is missed words “…doped with”.

=˃ corrected.

  • Line 98: The authors must explain or correct the strange order of commas in the molecular weights.

=˃ corrected

  • Line 102: The authors must specify the purity grades of the reagents that were used in the study.

=˃ purity of solvents are added

  • Line 105: The authors must specify the role of hydrothermal process in synthesis of TiO2and ZnO nanoparticles as the initial reagent also contained nanoparticles.

=˃ Added in the revised manuscript

  • Line 141: The authors should check model of the dielectric spectroscope. There is no “GMbH” model in website of Novo Control Technologies.

=˃ corrected

  • The first paragraph of section 3.2 describing composite fabrication as well as Figure 2 must be in section 2 where materials preparation is specified.

=˃ The suggested modification is done on the revised manuscript.

  • The authors must give detailed explanation of chemical reactions given in Figure 2.

=˃ The figure 2 does not show any chemical reactions as it is the schematic representation of the spinning set up. So we believe the reviewers mean figure 3, and a proper explanation is given in the section.

  • The authors must explain why flake-like nanoparticles of ZnO having size of several hundreds nm are not seen in SEM and TEM images given in Figure 3.

=˃During electrospinning the nanoparticles in small amounts get entangled with polymer chains and electrospinning can only cause the fiber formation. The nanoparticles are embedded within the polymer chains and this is the reason why solid fibers are only seen from electrospun fiber SEM.

  • Descriptions of Figure 3(f), Figure 4 (two bottom plots) are missed in the captions to these figures.

=˃ captions are modified

  • Line 221: The authors mention absorbance at 838 cm-1and 760 cm-1 in the text, but the same lines are marked as 840 cm-1 and 763 cm-1 in Figure 4. I think this discrepancy must be explained.

=˃ This error is rectified.

  • The peaks at 877 cm-11173 cm-1 have almost the same intensities for all the composites regardless there are or there are no inorganic nanoparticles. As the authors associate these peaks with β-phase and ZnO interaction with polymer chains, it must be explained why the undoped PH-PF possess the same magnitude and shape of these peaks.

=˃ FTIR image is provided with better resolution of peaks.

  • The authors should explain why for hybrid coaxial nanofibers that contain at least two different polymers (PVDF-HFP and PVDF) we can see only one (for each composite) melting peak in DSC graphs.

=˃ The reason for the presence of single melting peak can be the overlapping of melting peaks associated with both PVDF and its copolymer PVDF-HFP.

  • The legend in Figure 5 (d) must be shifted in order not to hide graph lines.

=˃The figure is modified

  • The authors must specify, what of the investigates composites is related to Figure 5 (c) and (d).

=˃Analysis of the DSC curve at different rates shows that the alteration of the heating rate causes a strong effect on the results delivered by differential scanning calorimetry. With all other  parameters  remaining  constant  the  peak  height  and  width  increases  with  increasing  heating rate. Phase transitions only occur or can be separated from the melting process if lower heating rates are used. From the cooling  rate,  it is very obvious that the crystallization peak is shifted in positive temperature  direction  with  decreasing  cooling  rates  and  the  absolute  height  of  the  peak  is  also decreasing. Apparently the correct choice of the cooling rate is as important as it is for the heating rate for the analysis of crystallization processes.

  • The authors must give a detailed description how piezoelectric tissues shown in photos in Figure 7 were produced of the synthesized nanofiber composites.

=˃ Sample details are mentioned as inset details.

  • The authors must give a detailed description of electrical measurement procedures. First of all, the load resistance must be specified. Second, what equipment was used to register data of Figure 6 (if oscilloscope, please give model)? Third, how deformation (or other type of piezoelectric excitation) was done and what was the magnitude of this excitation? In the current version of the manuscript these electrical measurements are the weakest part of the study.

=˃ The resistance value is provided, including all instrument details in the experimental section.

  • Why in Figure 6 voltage signals at all frequencies have the same period in time domain? If these plots are only schematic, I feel there is no need to give them in the paper, the resonance frequency just as numerical value will be enough.

=˃ The figure 6 is not a schematic. The values are obtained from the piezoelectric set up from the signal express software and plotted in origin. The time domain is fixed for easy understanding of the reader.

  • The author must give information how power densities and output current values were measured and by what equipment.

=˃ The experimental details are updated and power density equation is added in the revised manuscript.

  • Graphs in Figure 7 (c) and (d) are too regular for simple hand pressing and bending motions, the authors must explain it.

=˃ The real experiments of bending and pressing were done for longer time periods (2-3 minutes), and irregular results were observed. However here graphs are plotted within 4 seconds and that’s the reason why the values are regular.

  • Figure 8: Authors must specify what temperature was at measurements in (a) and (b) plots and what frequency was used for measuring (c) and (d) plots.

=˃ The values are provided.

  • There are no sudden changes typical for ferroelectric phase transition at 81°C in Figure 8. It seems to be that the transition is at temperatures out of given in Figure 8 range (maybe, near melting point, or what is referred to the melting in DSC graphs is phase transition?). The authors should explain their statement.

=˃ The temperature mentioned as 81 ºC is the Curie temperature. Form the dielectric data, the energy storage, relaxation time etc. are derived and correlated with other characterization. DSC is used to explain the crystallinity and transition temperature.

Round 2

Reviewer 1 Report

The authors have addressed my concerns in the revised manuscript. Therefore I recommend the revised manuscript for publication in Polymers journal.

Author Response

Thank you very much for the encouraging comments. 

Reviewer 2 Report

  • It seems to be there is missed words “…doped with” – inaccuracy is not corrected.
  • What composite is described in Figure 5 (c) and (d) – there is no data in caption or text.
  • It will be more informative to redraw Figure 6 as dependence of voltage on frequency
  • I am not satisfied by explanation about strange regular Figures 7 (c) and (d), the plots seem to be similar, despite different type of impact (pressing or bending). The detailed explanation should be added in the manuscript. If the authors mean peak values of voltage at sudden bending or pressing, they should also provide real raw data at least in the supporting information.
  • It is definitely not proven that 81C is the Curie temperature as the “sudden change in the dielectric constant” can be shifted left and right in the Figure 8 (c). Moreover, I do not understand what kind of magneto dielectric coupling is mentioned: first, the materials investigated in Ref. 48 are inorganic and differ from the PVDF and its copolymers even doped with nanoparticles, and second, nowhere in the reviewed manuscript the authors described any magnetic properties of their composites. The detailed analysis of this point must be added.

Author Response

  • It seems to be there is missed words “…doped with” – inaccuracy is not corrected.

We believe the reviewers meant to modify the sentence (line 52 in the original paper) as “doping with nanoparticles such as….”. The sentence is corrected as such.

  • What composite is described in Figure 5 (c) and (d) – there is no data in caption or text.

The sample analyzed was PHT-PFZ hybrid fiber, the information of which is added in the manuscript.

  • It will be more informative to redraw Figure 6 as dependence of voltage on frequency

We did not understand what the reviewer is exactly looking for. The origin data plots for the graphs are provided as supporting information for reviewers, for checking the data.

We followed similar plots in all other published research papers (reference papers are provided)

https://www.nature.com/articles/s41598-017-19082-3

https://www.sciencedirect.com/science/article/pii/S0264127519306148

  • I am not satisfied by explanation about strange regular Figures 7 (c) and (d), the plots seem to be similar, despite different type of impact (pressing or bending). The detailed explanation should be added in the manuscript. If the authors mean peak values of voltage at sudden bending or pressing, they should also provide real raw data at least in the supporting information.

Please note that the time given in the x-axis is in seconds, and the changes were observed in each tapping or bending process. Explanation is added in the revised manuscript, and the origin files are provided as supplementary information (reviewer can access the data).

  • It is definitely not proven that 81C is the Curie temperature as the “sudden change in the dielectric constant” can be shifted left and right in the Figure 8 (c). Moreover, I do not understand what kind of magneto dielectric coupling is mentioned: first, the materials investigated in Ref. 48 are inorganic and differ from the PVDF and its copolymers even doped with nanoparticles, and second, nowhere in the reviewed manuscript the authors described any magnetic properties of their composites. The detailed analysis of this point must be added.

The sentence is corrected and new reference is added.

Round 3

Reviewer 2 Report

  • Maybe I did not make myself clear. I mean as you show several plots of vibrational voltages having different magnitudes at their frequency it will be more descriptive for readers to give voltage-frequency plots, see e. g. Fig. 4 in https://doi.org/10.4028/www.scientific.net/AMM.761.579 or Fig. 2 in https://doi.org/10.3390/s19030614
  • As it follows from the manuscript: “The pressing and bending cycles show regular patterns as the 326 graphs represent very short time period. Noises were fully avoided for plotting the graphs.”. Do I truly understand you show some kind of schematic plots in Fig. 7(c) and (d)? If the answer is “yes”, you please write it distinctly in the manuscript so that to not confuse readers. Otherwise, you need to explain why and how you removed noise, add information about regular shape of the plots, describe the maximum magnitude of ca. 3 V and briefly explain why you have such slow response to short-time bending or pressing. I assure you, these results are interesting but presentation can be improved. It is bottleneck of the entire paper.